# LATENT TOPOLOGY INDUCTION FOR UNDERSTANDING CONTEXTUALIZED REPRESENTATIONS

## ABSTRACT

Recently, there has been considerable interests in understanding pretrained language models. This work studies the hidden geometry of the representation space of language models from a unique topological perspective. We hypothesize that there exist a network of latent anchor states summarizing the topology (neighbors and connectivity) of the representation space. we infer this latent network in a fully unsupervised way using a structured variational autoencoder. We show that such network exists in pretrained representations, but not in baseline random or positional embeddings. We connect the discovered topological structure to their linguistic interpretations. In this latent network, leave nodes can be grounded to word surface forms, anchor states can be grounded to linguistic categories, and connections between nodes and states can be grounded to phrase constructions and syntactic templates. We further show how such network evolves as the embeddings become more contextualized, with observational and statistical evidence demonstrating how contextualization helps words "receive meaning" from their topological neighbors via the anchor states. We demonstrate these insights with extensive experiments and visualizations.

## 1 INTRODUCTION

Recently, there has been large interests in analyzing pretrained language models (PLMs) (Rogers et al., 2020; Hewitt & Manning, 2019; Hewitt & Liang, 2019; Chen et al., 2021; Chi et al., 2020; Liu et al., 2019) due to their huge success. This work aims to investigate the topological properties, i.e., neighbors and connections of embeddings, of contextualized representations. Informally, we ask what does the "shape" of the representation manifold "look like", and what do they mean from a linguistic perspective. Formally, we hypothesize that there exists a spectrum of latent anchor embeddings serve as local topological centers within the manifold. As a quick first impression, Fig. 1 shows the latent states that we will discover in the following sections. Since such structure cannot be straightforwardly observed, we use unsupervised methods to infer the topology as latent variables.

Our unique topological perspective, combined with unsupervised latent variable induction technique, offers a systematically different methodology than the mainstream probing work. Most existing approaches usually define a *supervised* linear classifier as the probe (Hewitt & Manning, 2019; Hewitt & Liang, 2019; Hewitt et al., 2021; Liu et al., 2019), targeting for *pre-defined* properties using *pre-annotated* data. Such *a priori* approaches make *maximal pre-assumptions* and consequently, it would be hard to make new discoveries other than those are already assumed from the very beginning. Our work takes an *a posteriori* approach, which makes *mininal pre-assumptions* without using any annotation for supervision. Consequently, we achieve systematically different (yet complementary) results to the results from supervised probing literature. For example, while arguments make by supervised probing are strictly aspect-specific (e.g., how specific properties like syntax can be extracted out from other properties), our discoveries are more holistic and integrated (e.g., in Fig 1, we visualize all local topological centers as latent states, ground their meaning to lexical, syntactical, and semantic interpretations, and show how these properties are mixed with each other).

We use a structured variational autoencoder (VAE) (Diederik P. Kingma, 2013) to infer the latent topology, as VAEs are common and intuitive models for learning latent variables. We focus on the manifold where contextualized embeddings lay in (e.g., the last layer outputs of a fixed, not fine-tuned, BERT Devlin et al., 2019). We hypothesize there exists a wide spectrum of *static* latent states within

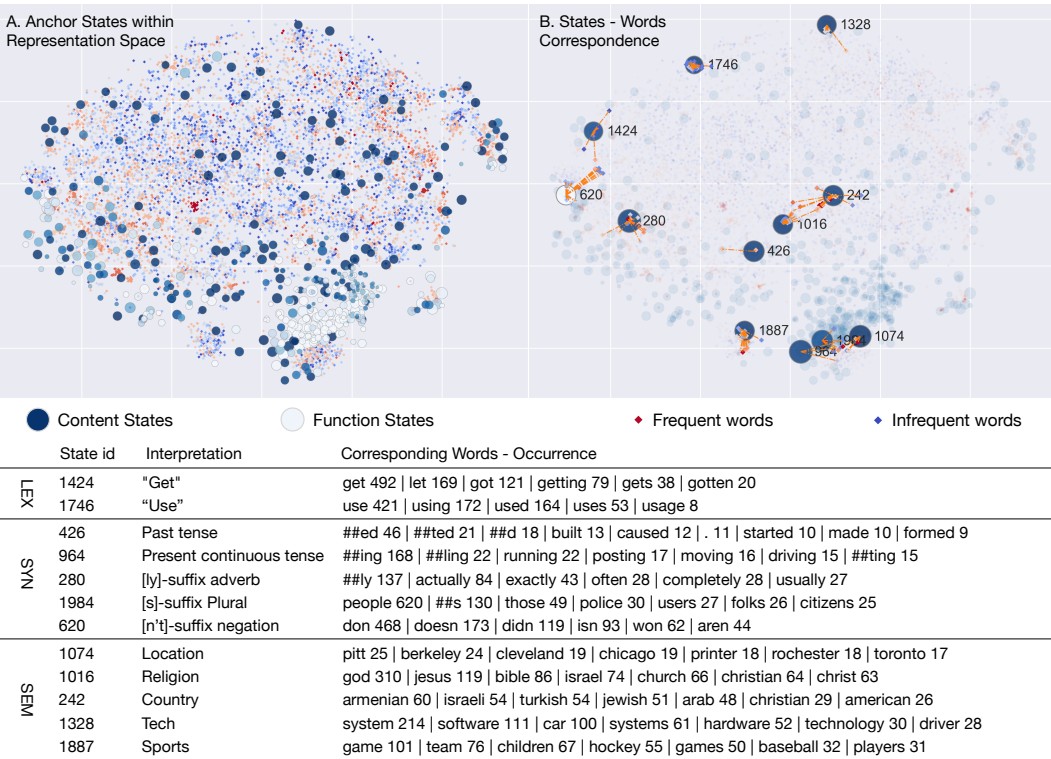

| | State id | Interpretation | Corresponding Words - Occurrence |
|---|---|---|---|
| LEX | 1424 | "Get" | get 492 \| let 169 \| got 121 \| getting 79 \| gets 38 \| gotten 20 |
| | 1746 | "Use" | use 421 \| using 172 \| used 164 \| uses 53 \| usage 8 |
| SYN | 426 | Past tense | ##ed 46 \| ##ted 21 \| ##d 18 \| built 13 \| caused 12 \| . 11 \| started 10 \| made 10 \| formed 9 |
| | 964 | Present continuous tense | ##ing 168 \| ##ling 22 \| running 22 \| posting 17 \| moving 16 \| driving 15 \| ##ting 15 |
| | 280 | [ly]-suffix adverb | ##ly 137 \| actually 84 \| exactly 43 \| often 28 \| completely 28 \| usually 27 |
| | 1984 | [s]-suffix Plural | people 620 \| ##s 130 \| those 49 \| police 30 \| users 27 \| folks 26 \| citizens 25 |
| | 620 | [n't]-suffix negation | don 468 \| doesn 173 \| didn 119 \| isn 93 \| won 62 \| aren 44 |
| SEM | 1074 | Location | pitt 25 \| berkeley 24 \| cleveland 19 \| chicago 19 \| printer 18 \| rochester 18 \| toronto 17 |
| | 1016 | Religion | god 310 \| jesus 119 \| bible 86 \| israel 74 \| church 66 \| christian 64 \| christ 63 |
| | 242 | Country | armenian 60 \| israeli 54 \| turkish 54 \| jewish 51 \| arab 48 \| christian 29 \| american 26 |
| | 1328 | Tech | system 214 \| software 111 \| car 100 \| systems 61 \| hardware 52 \| technology 30 \| driver 28 |
| | 1887 | Sports | game 101 \| team 76 \| children 67 \| hockey 55 \| games 50 \| baseball 32 \| players 31 |

Figure 1: A: There exist a spectrum of latent anchor states spread over the representation space serving as local topological centers. B and lower table: example linguistic interpretations and corresponding word distributions of latent states. Latent states encode a rich mixture of lexical, morphological, syntactic and semantic constructions.

this manifold and assume two basic and minimal generative properties of the states: (1). a state should summarize the meaning of its corresponding words and contexts; (2). transitions between different states should be able to reconstruct sentence structures. We model these two properties as emission and transition potentials of a CRF (Sutton et al., 2012) inference network. Since a VAE is a generative model trained by reconstructing sentences, essentially we infer states that are informative enough to generate/ reconstruct words and sentences.

In our experiments, we first show that the hypothesized topology does exist in embeddings produced by pretrained language models, but not in baseline (random and positional) embeddings. We further ground the discovered topology, i.e., the *local state-word emission* and the *global state-state transition*, to their linguistic interpretations. For local state-word emission, we show that states summarize rich types word surface forms ranging from lexicon, morphology, syntax to semantics variations. For global state-state transition, we differentiate two types of states within the space: states encoding function words and states encoding content words. We identify function states that serve as "hubs" in state transitions and attract content words of similar meanings to be close. Consequently, a rich set of phrase construction phenomena can be reconstructed from state transitions.

Furthermore, we highlight one important finding about how contextualization *directly changes the underlying topological structure*. We provide statistical and observational evidence. Statistically, we show that before contextualization, most function words are concentrated around a few head states and are isolated from content words; after contextualization, these function words spread over full state distribution and are better mixed with content words. Observationally, we see word neighbor structures change with contextualization. For example, before contextualization, the neighbors of suffix *#ed, #ing* are just random tokens; after contextualization, the neighbors of *#ed* become past tensed words like *had, did* and *used*, and the neighbors of *#ing* become present continuous tensed words like *running, processing* and *writing*.

Putting everything together, our results provide a holistic and analytical view of the representation space topology. In our discovered latent network, leave nodes can be grounded to word surface forms, anchor states can be grounded to linguistic categories, and connections between nodes and states can be grounded to phrase constructions and syntactic templates. We hope our work can inspire new insights and deeper understandings of pretrained language models.

## 2 RELATED WORK

**Geometric Properties of Language Models**     We are interested in the geometrics properties of contextualized embeddings (e.g., last layer output of BERT), which roughly asks what is the "shape" of the space and what they represent. Generally, an accurate and comprehensive geometric description of the manifold is challenging, due to its high-dimentional and non-linear nature. Previous work focuses on different geometric aspects, such as distance metrics (Wang & Ponce, 2021; Reif et al., 2019), linear subspaces (Hewitt & Manning, 2019; Cai et al., 2021), and hyperbolic geometry (Chen et al., 2021). We focus on the topological structure, which studies how embeddings and states are connected to each other thus forming meaningful latent networks.

**Relation to Supervised Probing**     Collectively known as "Bertology" (Hewitt & Manning, 2019; Rogers et al., 2020), the goal of probing is to discover what meaningful information are encoded within large language models. The mainstream approach is mostly *a priori*, which starts from a target linguistic property like part-of-speech (Liu et al., 2019; Hewitt et al., 2021), dependency edge (Tenney et al., 2019b;a), syntax trees (Hewitt & Manning, 2019; Hewitt & Liang, 2019; Reif et al., 2019; Chi et al., 2020), or sentiments (Chen et al., 2021; Wu et al., 2020), then fine-tune a supervised weak classifier (a.k.a. a probe). The higher the classification score, the more information contained. Our unsupervised approach takes a systematically different (and complementary) methodology from three perspectives: (1). our method is *posteriori* (v.s. existing work is *a priori*), and we hold *minimal assumptions* as we do not require supervision data, v.s., existing work holds *maximal pre-assumption* by targeting some property at the very beginning; (2). from an information theoretical perspective, existing work studies how to extract information from a channel (the probe) with limited capacity (Pimentel et al., 2020; Voita & Titov, 2020), while we directly study the original representation, without extracting information through a channel; (3). consequently, we show how different information is mixed with each other in the raw representation space, while existing work studies how one type of information can be extracted from the rest.

**Other Unsupervised Methods**     There are also unsupervised works proposes to extract syntactic (Kim et al., 2020), geometric (Cai et al., 2021), cluster-based (Dalvi et al., 2022), and other information from PLMs (Wu et al., 2020; Michael et al., 2020). Our focus here is the network topology within the representation space, which is not yet thoroughly studied. Amongst the large volume of Bertology research, the closest unsupervised work to ours are: Dalvi et al. (2022) who use clustering to discover latent concepts within BERT, and we will later use their results as a comparison to our discoveries; Michael et al. (2020) who discovers latent ontology in an unsupervised way; Cai et al. (2021) who study the geometric properties of BERT with a focus on isotropy. There is also supervised method for extracting static embeddings from contextualized embeddings (Gupta & Jaggi, 2021). These work more or less involve cluster structures within BERT. Our major difference is that we take an important further step from word clusters to state-state transitions and show how traversal over states leads to sentence constructions. In the latent variable literature, our inference model uses a classical CRF-VAE formulation (Fu et al., 2020; Mensch & Blondel, 2018). Existing work uses this formation for other tasks like structured prediction (Ammar et al., 2014) or sentence generation (Li & Rush, 2020) while we discover latent structures within PLMs.

## 3 METHOD

**Latent States within Representation Space**     Given a sentence $x = [x_1, ..., x_T]$, we denote its contextualized representations as $[r_1, ..., r_T] = \text{PLM}(x)$ where $\text{PLM}(\cdot)$ denotes a pretrained encoder (here we use BERT and our method is applicable to any PLM). Representations $r$ for all sentences lie in one manifold $\mathcal{M}$, namely the representation space of the language model. We hypothesize there exists a set of $N$ static latent states $s_1, ..., s_N$ that function as anchors and outline the space topology (recall Fig. 1). We emphasize that all parameters of the PLM are fixed (i.e., no fine-tuning), so all

learned states are intrinsically within $\mathcal{M}$. We focus on two topological relations: (1). state-word relations, which represent how word embeddings may be summarized by their states and how states can be explained by their corresponding words; (2). state-state relations, which capture how states interact with each other and how their transitions denote meaningful word combinations. Taken together, these two relations form a latent network within $\mathcal{M}$ (Fig. 1 and later Fig. 3).

**Modeling**    For state-word relations, we associate each word embedding $\boldsymbol{r}_t$ with a latent state indexed by $z_t \in \{1, ..., N\}$. We use an emission potential $\phi(x_t, z_t)$ to model how $z_t$ is likely to summarize $x_t$. The corresponding embedding of $z_t$ is then $\boldsymbol{s}_{z_t}$. For state-state relations, we assume a transition matrix $\Phi(z_{t-1}, z_t)$ modeling the affinity about how state $z_{t-1}$ are likely to transit to state $z_t$. Together $\phi(x_t, z_t)$ and $\Phi(z_{t-1}, z_t)$ form the potentials of a linear-chain CRF:

$$\log \phi(x_t, z_t) = \boldsymbol{r}_t^{\mathsf{T}} \boldsymbol{s}_{z_t} \qquad\qquad \log \Phi(z_{t-1}, z_t) = \boldsymbol{s}_{z_{t-1}}^{\mathsf{T}} \boldsymbol{s}_{z_t} \tag{1}$$

where the vector dot product follows the common practice of fine-tuning contextualized representations. The probability of a state sequence given a sentence is:

$$q_\psi(\boldsymbol{z}|\boldsymbol{x}) = \prod_{t=1}^{T} \Phi(z_{t-1}, z_t)\phi(x_t, z_t)/Z \tag{2}$$

where $Z$ is the partition function. Note that only embeddings of states $\psi = [\boldsymbol{s}_1, ..., \boldsymbol{s}_N]$ are learnable parameters of the inference model $q_\psi$. To infer $\boldsymbol{s}$, we use a common CRF-VAE architecture shared by previous work (Ammar et al., 2014; Li & Rush, 2020; Fu et al., 2022) and add a generative model on top of the encoder:

$$p_\theta(\boldsymbol{x}, \boldsymbol{z}) = \prod_t p(x_t|z_{1:t}) \cdot p(z_t|z_{1:t-1}) \qquad \boldsymbol{h}_t = \mathrm{Dec}(\boldsymbol{s}_{z_{t-1}}, \boldsymbol{h}_{t-1}) \tag{3}$$

$$p(z_t|z_{1:t-1}) = \mathrm{softmax}(\mathrm{FF}([\boldsymbol{s}_{z_t}; \boldsymbol{h}_t])) \qquad p(x_t|z_{1:t}) = \mathrm{softmax}(\mathrm{FF}(\boldsymbol{h}_t)) \tag{4}$$

where $\theta$ denotes the decoder parameters, $\mathrm{Dec}(\cdot)$ denotes the decoder (we use an LSTM), $\boldsymbol{h}_t$ denotes decoder states, and $\mathrm{FF}(\cdot)$ denotes a feed-forward network. We optimize the $\beta$-ELBO objective:

$$\mathcal{L}_{\mathrm{ELBO}} = \mathbb{E}_{q_\psi(z|x)}[\log p_\theta(x, z)] - \beta\mathcal{H}(q_\psi(z|x)) \tag{5}$$

We further note that the decoder's goal is for help inducing the latent states, rather than being a powerful sentence generator. After training, we simply drop the decoder and only look at the inferred states. Maximizing $p(x_t|z_{1:t})$ encourages $z_{1:t}$ (thus their embeddings $s_z$) to reconstruct the sentence and $p(z_t|z_{1:t-1})$ encourages previous $z_{1:t-1}$ to be predictive to the next $z_t$ (so we can learn transitions). Essentially, this formulation trys to find "generative" states $\boldsymbol{s}$ within the representation space $\mathcal{M}$ that are able to predict sentences and their next states.

## 4    EXPERIMENTAL SETTING

**Dataset, Model Parameters, and Learning**    We perform experiments on the 20News dataset (Kusner et al., 2015), a common dataset for latent variable modeling (initially for topic modeling) as our testbed. We primarily focus on the last layer output of a `BERT-base-uncased` model (BERTLAST), yet our method is applicable to any larger size, GPT-styled, or encoder-decoder-styled models. In terms of model parameters, the dimension of the states is the same as contextualized embeddings, which is 768. We use a light parameterization of the decoder and set its hidden state dimension to 200. Again, the purpose of the decoder is to help induce the states, rather than being a powerful sentence generator. We set the number of latent states $N$ to 2000. Recall that the vocabulary size of uncased BERT is 30522, which means that if uniform each state approximately corresponds to 15 words, serving as a type of "meta" word. We further note that setting $N = 2000$ is somehow a sweet point according to our initial experiments: larger $N$ (say 10K) is too fine-grained and under-clusters (words of similar linguistic roles are divided into different states) while smaller $N$ (say 100) over-clusters (words of different roles are gathered into the same cluster). Gradient-based learning of CRFs inference model is challenging due to the intermediate discrete structures. So we use approximate gradient and entropy from Fu et al. (2022) which enables memory-efficient differentiable training of our model. During training, we tune the $\beta$ parameter in Eq. 5 to prevent posterior collapse, which is a standard training technique for VAEs. All experiments are performed on Nvidia 2080Ti GPUs.

Table 1: Grounding inferred states to existing linguistic categories. POS and ENT can be inferred from the word directly while DEP , CCG and BCN require more context. Meaningful states emerge in BERTZERO and BERTLAST, but not in baseline POSITIONAL and RANDEMB.

| | POS | ENT | DEP | CCG | BCN |
|---|---|---|---|---|---|
| POSITIONAL | 3 | 0 | 3 | 1 | 1 |
| RANDEMB | 392 | 21 | 366 | 241 | 334 |
| BERTZERO | 673 | 37 | 468 | 450 | 275 |
| BERTLAST | 804 | 51 | 740 | 583 | 628 |

Table 2: Human evaluation (averaged over 3 annotators) of states not grounded to existing annotations. Although some states cannot be aligned with existing linguistic annotations, most of them are still meaningful to the annotators and only a small portion of them are truly uninterpretable.

| | LEX | SYN | SEM | Not Interpretable |
|---|---|---|---|---|
| BERTZERO | 83 | 54 | 135 | 39 |
| BERTLAST | 26 | 40 | 56 | 22 |

**Baseline Embeddings** We compare against: (1). POSITIONAL states induced from positional embeddings. As there is no content information, we expect the induced structures to be very poor. (2). RANDEMB, fixed random state embeddings sampled from a Gaussian distribution sharing the same mean and variance with BERTLAST embeddings. We also expect no meaningful topology from this embedding. (3). BERTZERO, static word embeddings from the zeroth layer of BERT. We expect linguistically meaningful topology emerge within BERTZERO and BERTLAST. Furthermore, we are particularly interested in the comparison between BERTZERO and BERTLAST, as the differences can shed light on what happens after contextualization.

## 5 DISCOVERIES

The first part of our experiments studies on *local topology* of the representation space, focusing on how states serve as topological centers of their corresponding words. we first show that states from language models can either be grounded to existing annotations like POS tags, or can be recognized by human annotators as a lexical, syntactic, or semantic category, while states from baseline embeddings cannot be grounded. Then show how contextualization shapes the space topology by comparing BERTZERO and BERTLAST, highlighting how state-word connections change with contextualizaiton.

The second part of our experiments focus on *global topology* of the representation space, highlighting how transitions between states form the backbone of the overall space structure. We identify important function states that serve as hubs of the latent network, attracting content words of similar meaning to be topologically close to each other. We also show how contextualization shapes the global transitions. Finally, we put everything together and give a holistic statement of the space structure and their linguistic interpretations.

### 5.1 LOCAL STATE-WORD TOPOLOGY

**Emergence of Linguistically Meaningful States** To see whether the inferred states are meaningful, we ground them to common linguistic categories. Specifically, we consider: (1). POS, part of speech tags; (2). ENT, named entities; (3). DEP, labels of dependency edges linking a word to its dependency head; (4). CCG, CCG supertags which contain rich syntax tree information and are usually referred as "almost parsing"(Liu et al., 2019); (5). BCN, BERT Concept Net from Dalvi et al. (2022), a hierarchy of concepts induced from BERT, mostly about semantics and similar to a topic model. We obtain the POS, ENT, DEP annotations on our 20News dataset using a pretrained parser

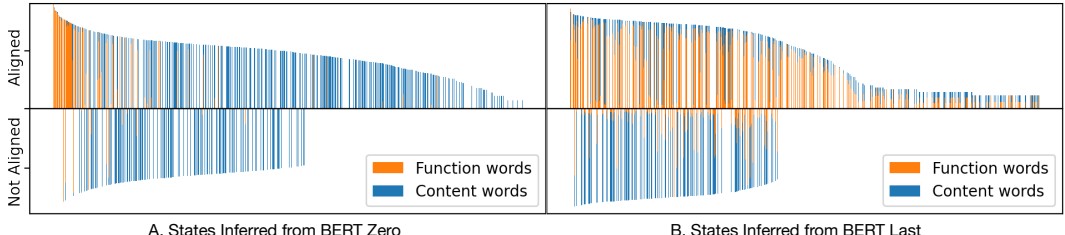

Figure 2: Contextualization changes space topology. Each bar represents a latent state with height equals to log frequency. Orange/ blue represents the portion of function/ content word corresponding to a state. In BERTZERO, most function words are concentrated around head states. After contextualization (BERTLAST), function words mix with content words and spread over all states.

with the `spacy` library. We obtain the CCG and BCN annotations from their own websites. After training, we use Viterbi decoding to decode states, and each state may correspond to multiple copies of the same word from different contexts. We say a latent state aligns with a predefined tag if 90% of the word occurrence corresponds to this state also corresponds to a tag. For example, suppose state-0 occurs 100 times in the full validation set, 90 times of which correspond to either "happy" or "sad" and 10 times correspond to other random words. In this case, "happy" and "sad" takes the dominant portion of state-0 (90 out of 100), and both are adjectives, so we say state-0 aligns with the POS tag adjective. We set the threshold to 90% because it is intuitively high enough. We select the model that has the largest number of aligned tags to the union of the five types of annotations during validation.

Table 1 shows the grounding results. We see there are significantly less number of states in PO-SITIONAL and RANDEMB that can be grounded to existing categories, and states inferred from BERTZERO and BERTLAST can be better grounded (e.g, 804 states in BERTLAST can be grounded to POS tags, while this number in RANDEMB is only 392). These results mean that meaningful states emerges in embeddings from language models, but not in baseline embeddings. Note that although there are still certain numbers of states from RANDEMB that can be grounded, it can be explained by the fact that the training process still tries to place states that are useful for sentence resonstruction, even these states are placed between random embeddings. We also notice that states from BERTLAST aligns better with categories requiring more context (DEP, CCG and BCN), this is also expected as BERTLAST is contextualized while BERTZERO is not.

For states that cannot be grounded to existing categories, we show that they are still linguistically meaningful by human annotation. Specifically, we ask human annotators (three graduate students with 100+ TOEFL test scores) to annotate if the corresponding words of a state are: (1). LEX: words that are textually similar. (2). SYN: words share similar morphological-syntactic rules. (3). SEM: words with related meaning. (4). N.I.: Not Interpretable. The results are show in Table 2. Generally, most of the states still contain meaningful linguistic information from the annotators perspective and only a small portion of them are truly uninterpretable.

**Contextualization Shapes State Topology**     Now we show how contextualization shapes the space topology by taking a closer look on what is encoded in BERTLAST and *changes in* BERTZERO. To this end, we differentiate two types of words: (1). function words (e.g., preposition, conjunction, determiner, punctuation .etc) whose main role is to help sentence construction but do not have concrete meanings on their own; (2). content words (e.g., nouns, adjectives, verbs, adverbs) who have concrete meaning. It turns out that contextualization results in very different behavior about the encoding of these two types.

Figure 2 shows how function/ content words are encoded before/ after contextualization. We see two effects of contextualization: (1) before contextualization, most function words are concentrated around a few head states; after contextualization, these function words spread over the full distribution, not just head states. This shows that the meaning of function words is distributed from head states to all states according to their context. (2). before contextualization, most states are either function-only or content-only (as most bars are either orange-only or blue-only); after contextualization, most states contain both function and content states (as most bars have both blue and orange portions). This shows that the meaning of function words is entangled together with their neighbor content words. Intuitively, contextualization helps function words "receive" meaning from their context.

Table 3: Contextualization changes local neighbor structures. Try comparing red tokens and see their neighbor words before/ after contextualization. Tokens previously opaque (as their corresponding latent state are not meaningful) gain linguistic clarity (as their corresponding states encode meaningful linguistic constructions).

| | Before Contextualization (BERTZERO) | After Contextualization (BERTLAST) |
|---|---|---|
| Symbol | $\$_{2851}$ \| size$_{56}$ \| type$_{49}$ \| numbers$_{38}$ \| number$_{35}$ | $\$_{248}$ \| money$_{76}$ \| cost$_{64}$ \| pay$_{54}$ \| love$_{42}$ \| worth$_{41}$ |
| | $@_{13522}$ \| same$_{2110}$ \| ordinary$_{46}$ \| average$_7$ | $@_{1184}$ \| com$_{1146}$ \| org$_{232}$ \| address$_{91}$ \| list$_{75}$ |
| Prefix | re$_{5635}$ \| pre$_{559}$ \| mis$_{481}$ \| co$_{258}$ \| pr$_{22}$ | old$_{657}$ \| after$_{42}$ \| recently$_{42}$ \| years$_{36}$ \| pre$_{36}$ |
| | un$_{1922}$ \| per$_{871}$ \| di$_{468}$ \| multi$_{237}$ \| #con$_{159}$ | un$_{1524}$ \| in$_{562}$ \| im$_{275}$ \| mis$_{162}$ \| con$_{155}$ \| um$_{148}$ |
| Suffix | #ing$_{1508}$ \| #ting$_{108}$ \| #ley$_{56}$ \| #light$_{36}$ | #ing$_{1563}$ \| running$_{226}$ \| processing$_{118}$ \| writing$_{98}$ |
| | #ly$_{1722}$ \| dear$_{59}$ \| thy$_{36}$ \| #more$_{15}$ \| #rous$_9$ | #ly$_{983}$ \| actually$_{645}$ \| exactly$_{325}$ \| simply$_{282}$ |
| | #eg$_{404}$ \| #ed$_{385}$ \| #ve$_{189}$ \| #ize$_{183}$ \| #ig$_{164}$ \| | #d$_{1012}$ \| had$_{542}$ \| #ed$_{416}$ \| did$_{320}$ \| used$_{258}$ |
| | #s$_{348}$ \| #l$_{102}$ \| #t$_{98}$ \| #p$_{85}$ \| #m$_{64}$ \| #u$_{62}$ | #s$_{1839}$ \| files$_{225}$ \| books$_{169}$ \| machines$_{123}$ |
| | #s$_{335}$ \| s$_{333}$ \| #t$_{134}$ \| #p$_{120}$ \| #u$_{117}$ \| it$_{98}$ | people$_{4481}$ \| #s$_{682}$ \| those$_{361}$ \| users$_{210}$ \| folks$_{193}$ |
| Lexicon | decided$_{250}$ \| decision$_{211}$ \| decide$_{189}$ \| determine$_{102}$ \| determined$_{99}$ \| decisions$_{82}$ | decision$_{220}$ \| position$_{206}$ \| choose$_{155}$ \| command$_{147}$ \| actions$_{106}$ \| decide$_{94}$ |
| | be$_{7125}$ \| been$_{1591}$ \| being$_{257}$ \| gone$_1$ | be$_{7192}$ \| are$_{3099}$ \| am$_{1139}$ \| is$_{165}$ \| become$_{52}$ |

We now revisit Fig. 1 that we briefly mentioned in §1. Figure 1 is produced by t-SNE (Van der Maaten & Hinton, 2008) jointly over the states and embeddings from BERTLAST and illustrates the local topology (because t-SNE preserves more local information) of the representation space. Blue/ white circles in Fig. 1 correspond to blue/ orange bars in Fig. 2. It directly shows how states spread over and "receive" meaning from their neighbor word embeddings and encode to different types of word clusters.

We further highlight certain topological changes before/ after contextualization in Table 3. Before contextualization, we see (1). the symbol $ does not have meaningful neighbors; (2). the suffix *#ing* and *#ed* are just ordinary subwords; (3). the word *be*'s neighbor is its morphological variants. After contextualization, we see (1). the symbol $ encodes money; (2). the suffix *#ing* and *#ed* encode tense; (3). *be*'s neighbor becomes linking verbs. Contextualization makes these tokens "receive" meaning from their contexts.

## 5.2 GLOBAL STATE TRANSITION TOPOLOGY

Now we move our focus from local topology (how states form local centers of their neighbor words) to global topology (how all states are inter-connected to each other). We first visualize the transition network to see how backbone of the space topology follows a clear long-tail structure. We show state transitions encode meaningful phrase construction and contextualization changes the transition topology. Finally, we put everything together and give a holistic statement of the space topology.

**Overall State Transition Topology** We visualize the induced state-state network in Fig. 3 using t-SNE again (this time without word embeddings). Blue circles represent states with more content words while white circles represent states with more function words. Circle size represents state frequency. To see how states transit to each other, we compute the state transition statistics from the state sequences decoded from the validation dataset. The transition histogram is also shown in Fig. 4. We use blue edges to denote frequent (stronger) transitions and yellow edges to denote less frequent (weaker) transitions. Table 4 shows example transitions and their corresponding word bigram occurrences. In Fig. 3A we see: (1). both nodes and edges follow a *long-tail* distribution: there are few frequent nodes/ edges taking the head portion of the distribution, and many infrequent nodes/ edges taking the tail portion of the distribution. Note that the yellow background in Fig. 3A consists of many weak edges. (2). frequent states are more inter-connected and tail states are more spread. Fig. 3B zooms in the top states, and we see function states usually as the *hub* of the edges. This is also evidenced in Table 4, as we can see many different content words transit to the function word *to* (e.g., free-to, willing-to, go-to, went-to), and the function word "to" can transit to other content words (e.g., to-buy, to-sell, to-build). Here the state encoding *to* is a hub connecting other states and words.

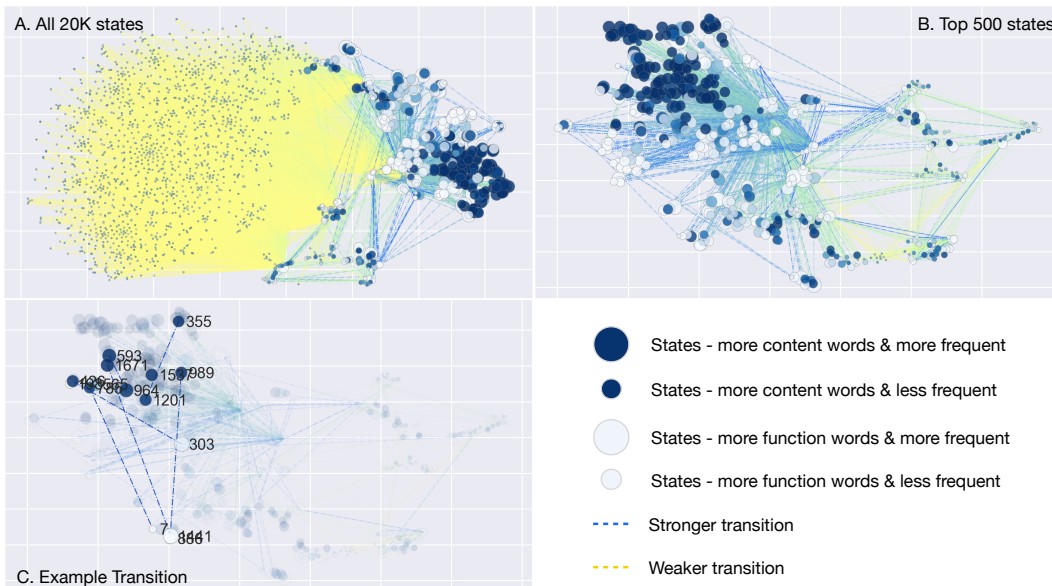

Figure 3: A: State transitions, as the backbone of the representation space, show a clear *long-tail* structure. Tail states are more spread and transitions to tail states mostly are from head states (yellow edges). Head states are more inter-connected (blue edges). Blue edges denote more frequent (stronger) transitions and yellow edges denote less frequent (weaker) transitions. B: transitions between top states, function states usually (white circles) serve as hubs of transitions (many white circles at the center). C. Example transitions, see Table 4 below for their interpretations.

Table 4: Example state transitions. Function words (like "to") serve as hubs that attract content words of similar meanings to be within the same latent state. Subscript numbers denote occurrence.

| Transition | Example states from BERTLAST | Explanation |
|---|---|---|
| Function word + content word | | |
| 886-989 | to-buy$_{21}$ \| to-sell$_{13}$ \| to-build$_5$ \| to-purchase$_5$ \| to-create$_4$ \| to-produce$_3$ | to do sth. |
| 1671-1441 | free-to$_{14}$ \| willing-to$_9$ \| hard-to$_6$ \| easy-to$_4$ \| happy-to$_3$ \| safe-to$_2$ | adjective + to |
| 785-7 | go-to$_{19}$ \| going-to$_{10}$ \| went-to$_5$ \| trip-to$_4$ \| moved-to$_3$ \| come-to$_2$ | movement + to |
| 785-565 | come-out$_9$ \| come-up$_6$ \| went-up$_4$ \| go-down$_3$ \| went-out$_3$ \| went-back$_1$ | move + direction |
| 426-198 | caused-by$_9$ \| #ed-by$_8$ \| #ted-by$_5$ \| produced-by$_4$ \| made-by$_4$ \| driven-by$_3$ | passive voice |
| 303-426 | is-made$_2$ \| be-converted$_2$ \| been-formed$_2$ \| are-formed$_2$ \| been-developed$_1$ | passive voice |
| Content word + content word | | |
| 1537-1537 | ms-windows$_{13}$ \| source-code$_5$ \| windows-program$_3$ \| operating-system$_3$ | computers |
| 355-1201 | three-years$_7$ \| five-years$_3$ \| 24-hours$_3$ \| ten-years$_2$ \| 21-days$_2$ | time |
| 593-964 | image-processing$_2$ \| meter-reading$_1$ \| missile-spotting$_1$ \| speed-scanning$_1$ | v.ing as noun |

Figure 4 shows transition distribution. The bars here correspond to edges in Fig. 3. Color denote the portion of function/ content words (node color in Fig. 3) and height correspond to edge color in Fig. 3. We observe: (1). transitions are usually mixtures of function and content states. (2). contextualization makes function words less concentrated around head transitions (in Fig. 4 left, BERTLAST has less orange portion than BERTZERO), and more spread within the distribution (in Fig. 4 right, BERTLAST has a longer tail of orange bars than BERTZERO).

## 5.3 A HOLISTIC STATEMENT OF LATENT SPACE TOPOLOGY

Now we can finally put everything together and reach a holistic and bottom-up description about how words, phrases, and sentences are constructed within the space topology. This mechanism consists of

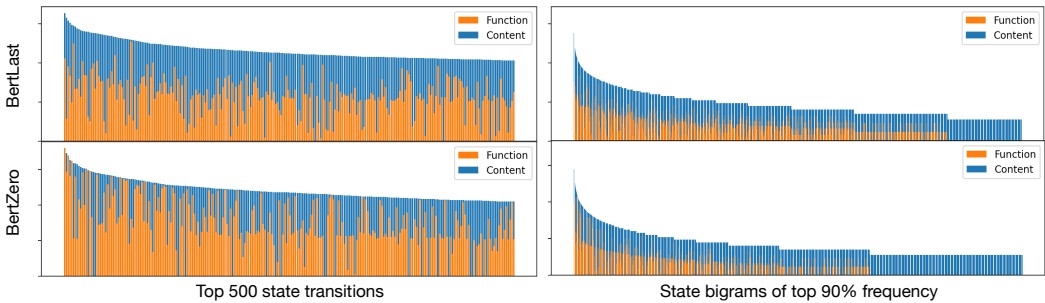

Figure 4: State transition distribution. Again, function words exist more at top transitions in BERTZERO. After contextualization (BERTLAST), they become more spread within the distribution.

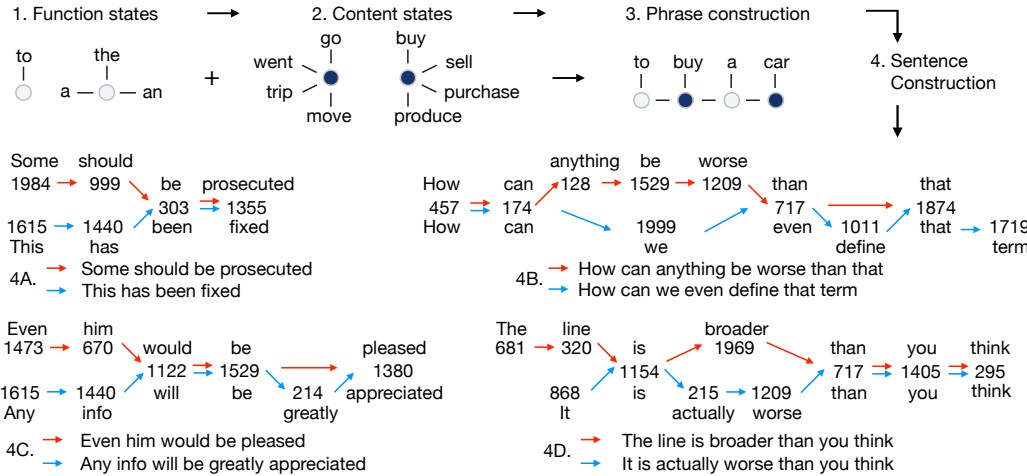

Figure 5: Four steps illustration of how words, phrases, and sentences are constructed as nodes, connections between nodes, and traversals over the latent topology. Numbers mean latent state index. Structurally similar sentences share overlapped paths of latent states.

four steps, and is illustrated in Fig.5. Step 1: there exist function states that correspond to specific function words (as is evidenced in Fig. 2). Step 2: there exist content states that correspond to content words with similar lexical/ syntactic/ semantic meaning (as is evidenced in Table 1 and Fig. 1). Step 3: transitions between function and content states correspond to meaningful phrase constructions (as is evidenced in Fig. 3 and 4, Table 4). Step 4: a traversal of states encodes a sentence within the space (this is a corollary combining step 1-3). Fig. 5 shows sentences sharing overlapped traversal. State transition chains encode the sentence templates and state-word emissions fill in the content. When the transition chains of two sentences overlap, the two sentences tend to be syntactically similar. Overall, in this latent network, leave nodes correspond to specific words, anchor states correspond to linguistic categories, and connections between nodes and states correspond to phrase constructions and syntactic templates.

## 6 CONCLUSIONS

In this work, we study the topological structure of the representation space of contextualized representations. Our analytics starts from the hypothesis that there exists a latent network of states that summarize the representation space topology. We verify such states exist in embeddings from language models, but not in basedline embeddings. We further study how state transitions mark the backbone of the representation space and encode meaningful phrase constructions. Contextualization shapes both local and global topology of the representation space. We hope this work deepens the understanding of language models and inspires new modeling techniques based on the topology of the representations.

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

## A    APPENDIX

You may include other additional sections here.

