# OpenReview forum: "Latent Topology Induction for Understanding Contextualized Representations"
_ICLR.cc/2023/Conference — Submitted to ICLR 2023_

### Official Review · Reviewer_mZMM · 2022-10-23

**Confidence:** 3
**Correctness:** 2
**Technical Novelty And Significance:** 3
**Empirical Novelty And Significance:** 2
**Recommendation:** 5

**Clarity, Quality, Novelty And Reproducibility:**

Unsupervised methods are an exciting new frontier to study BERTology. They do away from the supervision which requires annotated data and limited to the preconceived knowledge basis. The authors presented a novel unsupervised method to study pLMs.

I believe the evaluation could've been carried better by including more pLMs, by having probing methods as baseline.

I found the claims to be a bit strong and perhaps more validation is required.

**Strength And Weaknesses:**

Strengths:

Unsupervised methods are an exciting alternative to understanding pLMs as shown by some recent work (Micheal et al 2020, Dalvi et al 2022). The authors have presented a novel method along this line of work.

I liked the visualizations in the paper.

The idea to align CRF states (as a proxy to pLMs) with the linguistic concepts is interesting.

Weaknesses:

I would have liked the authors to study models other than BERT to strengthen their analysis and show if their conclusions are general across different models. They also only explored the initial and final layers of BERT. Prior analysis has shown that much of the linguistic knowledge is present in the middle layers. Initial layers are mostly reconstruction word structure from subword segmentation and final layers are mostly geared towards the task at hand.

While the method is exciting, there are not enough crisp findings in the paper. Results such as lower layers capture lexical concepts and higher layers learning contextual information is not a novel discovery.

BERT and likes are trained to predict the masked word. They are not known to learn sentence representation. It is therefore quite bold to assume that states can be traversed over the discovery typology to induce phrases and sentences. This requires further investigation, but the authors just flashed this towards the end and did not spend more time and experimentation towards this.

It would have been great to somehow use any probing method as a baseline to match and contrast the findings we obtain in that framework. Commenting on the strengths and weaknesses of both with empirical evident would make the paper much stronger.

I think the paper is not written well. It’s hard for a novice reader to understand.


**Summary Of The Paper:**

Summary

Representations learned within deep transformer models learn rich linguistic knowledge. Most of the prior work uses a probing framework which is limited to pre-defined aspects of language. Recent methods are using unsupervised methods to study ontology in pLMs. This paper aims at discovering latent states that can provide a generative explanation of the representations. The authors train a sequence-level CRF on top of contextualized vectors, which are then stacked with an LSTM and trained in an autoencoder fashion. The authors then align stats of the CRF with linguistic annotations.

The authors found that: i) the states don’t necessarily align with linguistic concepts, ii) contextualization causes function words to acquire meaning from the content words, iii) lower layers encode shallow lexical concepts while the higher layers account for contextual information. Finally the authors showed that the states of CRF can be traversed to discover meaningful phrasal constructions.




**Summary Of The Review:**

I believe this is a good paper under progress. The method is novel but not executed well. More exploration across different dimensions such as including more pLMs possibly including LSTMs and static models, exploring the whole network and not just 2 layers, including probing as baseline would make the paper stronger. This will hopefully bring more interesting findings and also put the authors on a better footings for the claims they are making.

---

### Official Review · Reviewer_QWYM · 2022-10-23

**Confidence:** 3
**Correctness:** 1
**Technical Novelty And Significance:** 2
**Empirical Novelty And Significance:** 2
**Recommendation:** 1

**Clarity, Quality, Novelty And Reproducibility:**

This work is very difficult for me to assess. In my opinion, there does not seem to be rigorous empirical evidence of the contributions the authors state in the paper. I think the method proposed is straightforward but am unable to come to the same conclusions as the authors with the limited evaluation, small scale studies and anecdotal findings.

**Strength And Weaknesses:**

Strengths:

- The authors provide nice illustrative figures for their methods

Weaknesses:

- To me it is unclear what types of claims can be made rigorously in this work and it seems like the authors have presented some interesting anecdotes but the evaluation methodology is not made explicit. Many of the claims including those relating to traversal uncovering phrases and sentences along with the study on states encoding linguistic information seem unfounded based on the limited experimentation in the paper.
- When reading the paper, I am unable to reconcile the bold claims made in the introduction with the presented empirical studies.
- Small scale human studies are shown with three individuals with similar background (e.g. graduate students) performing annotators. This seems like an especially small sample size with a lot of bias and the authors should recognize that a study run in this manner should not be used to bolster claims.
- It is unclear how selection of states to provide annotation for was proposed and the experiments, in my opinion, are not very conclusive due to the experimentation setting.
- The authors claim that N=2000 is a sweet point but provide no clear evidence for this other than some further anecdotes.

**Summary Of The Paper:**

In this paper, the authors study the topology of representations in language models including how contextualization affects neighbors and connectivity, relationships with linguistic interpretations, and evolution of embeddings through model layers. First the authors consider state-word emissions and global state-state transitions and show that states can summarize linguistic information such as lexicon, morphology, syntax and semantics. Next the authors highlight the effect of contextualization on the topological structure of representations.  In particular, they find that word neighbor structure changes with contextualization.

**Summary Of The Review:**

The authors present interesting ideas for understanding latent topology of the contextual representations. This is an important and challenging problem but the method and empirical evaluations are limited, hard to decipher, and not based on large scale evaluations. Many anecdotes are provided in the paper but it is difficult to objectively make the conclusions the authors arrive at in this work.

---

### Official Review · Reviewer_5qaG · 2022-10-25

**Confidence:** 3
**Correctness:** 3
**Technical Novelty And Significance:** 3
**Empirical Novelty And Significance:** 3
**Recommendation:** 6

**Clarity, Quality, Novelty And Reproducibility:**

The analysis presented in the paper is insightful and fairly comprehensive. The paper is reasonably clear, although abbreviations in some tables (e.g. 1, 2) should likely be described in the captions, rather than in the main text only. There are a few typos (e.g. P1 "we" -> "We", P1 "arguments make by", P5 "focus" -> "focuses", P9 "basedline") and I would suggest replacing "leave node" by "leaf node". As far as I know, the approach is novel, but I may not be familiar with all relevant work. General observations could likely be reproduced with some effort, but not all hyper-parameters are described.

Other/questions

In Eq. 4, it doesn't appear that $p(x_t|z_{1:t})$ actually depends on $z_t$. Is that correct?

It would have been interesting to compare how the topology differs between masked and autoregressive language models.

**Strength And Weaknesses:**

Strengths

The analysis is quite detailed, describing both interactions between states and tokens, as well as transitions across states to generate phrases.

The paper compares both local (token embeddings) and contextualized (last layer) representations, showing how function words become more spread out across states in the latter case.

Weaknesses

State transitions appear to be an artifact of the CRF-VAE, rather than an intrinsic property of the BERT representations.

There are quite a few linguistically grounded states for random embeddings (Table 1), although less than for BERT representations. While the paper briefly addresses this phenomena, it should be described more clearly. How could we know from the results only that there is no meaningful underlying topology?

**Summary Of The Paper:**

This paper exposes the hidden topology behind a pretrained BERT language model by training a CRF-VAE, which describes latent states and their relationships. It shows that many of the discovered states are linguistically meaningful. State transitions are also interpretable and can be followed to construct sentences.

**Summary Of The Review:**

This paper introduces an insightful approach to infer the topology of a pretrained language model. However, whether this structure (especially state transitions) is intrinsic to the underlying language model is somewhat ambiguous.

---

### Official Review · Reviewer_ZfD1 · 2022-10-30

**Confidence:** 4
**Correctness:** 3
**Technical Novelty And Significance:** 3
**Empirical Novelty And Significance:** 3
**Recommendation:** 5

**Clarity, Quality, Novelty And Reproducibility:**

There are a large number of typos and grammatical errors, some of which are captured below. Conceptually, the work is of high quality and novelty.

At a high-level, I am confident I could *generally* reproduce the architecture used to obtain the latent state embeddings, however I'm sure there are many important details in the exact implementation which were not included in the paper (nor in any appendix). It would be useful if the authors could provide their code to ensure reproducibility.

### Typos / Suggestions
p. 1: "we" -> We

p. 1: "such network" -> such a network

p. 1: "leave" -> leaf

p. 1: "such network" -> such a network

p. 1: "been large interests" -> been a large interest

p. 1: "do they" -> does it

p. 1: "serve" -> serving

p. 1: "those are" -> those that are

p. 1: "mininal" -> minimal

p. 1: "make" -> made

p. 1: "lay in" -> lie

p. 2: "rich types word surface forms" -> ?

p. 3: "leave" -> leaf

p. 3: "understandings" -> understanding

p. 3: "geometrics" -> geometric

p. 3: "with each other" -> together

p. 3: "proposes" -> proposed

p. 3: "supervised" -> a supervised

p. 3: "BERT and" -> BERT, but

p. 4: "how zt is likely" -> how likely state $z_t$ is

p. 5: "we" -> We

p. 5: "referred as" -> referred to as

p. 5: "BCN, BERT Concept Net" -> BERT Concept Net (BCN)

p. 6: "occurrence corresponds" -> occurrence corresponding

p. 6: "90 times" -> 90 instances

p. 6: "10 times" -> 10 instances

p. 6: "random" -> <delete>

p. 6: "states emerges" -> states emerge

p. 6: "resonstruction" -> reconstruction

p. 6: "even these states" -> even though these states

p. 6: "aligns better" -> align better

p. 6: ", this is also" -> , which is

p. 6: "punctuation .etc" -> punctuation, etc.

p. 6: "who" -> which

p. 7: "transit" -> transition

p. 9: "leave" -> leaf

p. 9: "basedline" -> baseline

**Strength And Weaknesses:**

### Strengths
1. The provided approach is straightforward and follows a generally useful high-level principle in ML - if the model you are using does not have a certain property (in this case, interpretability), simply fit a model which *does* have this property on top of your model.
2. The qualitative results shown are compelling, and seem to provide insight into otherwise opaque language models.

### Weaknesses
1. The biggest concern for this approach is whether the analysis is providing insight into the *embedding geometry* of BERT, or simply the statistics of the data itself, for which the embeddings of BERT are a very strong proxy. For example, it seems like it would be interesting to compare a CRF model trained directly on the surface tokens with the BERTZero analysis. I wonder if the authors had any thoughts on this.
2. Table 2: As explained, this evaluation seems highly susceptible to confirmation bias. To avoid this, it seems pertinent to (at least) include states from RandEmb and Positional versions as well, and perhaps some other external sources also (eg. synsets from WordNet, ConceptNet, alternative linguistic similarity datasets) to provide externally-validated "interpretable" and "non-interpretable" sets. Such an evaluation would be far more robust, though the same annotators who performed the current annotations in Table 2 would not be able to take part in this annotation, as they have already observed the clusters from BERTZero and BERTLast.
3. I was surprised that there was not a large appendix with an extensive collection of qualitative examples of the groundings for various latent states, as well as a critical analysis of when things went wrong. Even better would be an interactive online demo (a la exBERT) where one can observe the latent clusters and transitions for a given sentence. The absence of an extensive compilation of analyses is somewhat surprising for a paper of this style.
4. (Minor) The writing needs some work, there are a large number of grammatical and typographical errors (see the Clarity section).

### Questions

1. p.2 “For example, before contextualization, the neighbors of suffix #ed, #ing are just random tokens; after contextualization, the neighbors of #ed become past tensed words like had, did and used, and the neighbors of #ing become present continuous tensed words like running, processing and writing”  It is unsurprising that the word neighbor structures change with contextualization, however this particular example seems a bit surprising, because it would seem (a priori) that the suffixes #ed and #ing will always be surrounded by past and current-tense words, regardless of context. Is my intuition incorrect here?
2. p.4 “Again, the purpose of the decoder is to help induce the states, rather than being a powerful sentence generator.”  While this is true, it also seems like it might be useful to report some measure of quality of the decoder. There are various degenerate settings in which the results presented in Table 1 (or, indeed, even better) could be obtained but which do not actually capture anything about the topology. For example, if 1999 of the latent states were all used to represent a particular word (say "ball"), and the last latent state was used for every other word. Obviously this would be a terrible sentence predictor, but it would do very well on the evaluation in table 2. To some extent, the additional analysis (eg. Figure 2) provides reasonable evidence that this is not occurring. Just to clarify - the model evaluated in all later sections is the same as that which yielded the results presented in Table 1, correct?
3. Table 1:  Could you show how many elements were not match to any category? Based on Table 2, it looks like 311 for BERTZero and 144 for BERTLast, which implies that 218 states in the first case and  950 states in the second were mapped to more than one category. It would be interesting to show some analysis or examples of states which were classified in multiple categories.

**Summary Of The Paper:**

This paper introduces a novel method for inferring structure present in the latent embedding layers of large language models. The proposed method predicts word embeddings from latent states and state transitions in the form of a linear CRF. These states conceptually represent "meta" words, and the authors use a CRF-VAE architecture to learn the latent states. They analyze the resulting latent state embedding by attempting to identify how often a given latent state will produce a word in a consistent "category", where the categories include things such as part of speech and dependency edges. They compare this analysis on both random and pretrained BERT models and observe that the latent states trained on the pretrained BERT model have a higher tendency to capture meaningful linguistic categories. They further include analysis on how this changes between the initial embeddings and contextualized embeddings (last layer) of BERT, concluding that there are "anchor states" which correspond to linguistic categories, a long-tail of "leaf nodes" which correspond to specific words, and that the state transitions represent meaningful phrases.

**Summary Of The Review:**

This work has a novel approach to interpreting the structure present in large language models. While the current presentation is a bit rough, I did not find any glaring conceptual errors, and I am sure that the authors will be able to improve things for a potential camera-ready version. I think the work is potentially worthy of acceptance, pending answers to some of the questions I posed above in the "weaknesses" section.

---

### Decision · Program_Chairs · 2023-01-20

**Decision:**

Reject

**Justification For Why Not Higher Score:**

Need more empirical validations to support the claims.

**Justification For Why Not Lower Score:**

N/A

**Metareview: Summary, Strengths And Weaknesses:**

This paper analyzes the representation space of language models from a topological perspective and connects it to their linguistic interpretations. I think this is an interesting research topic. However, the reviewers agree that more thorough empirical validations are needed to support the claims that are made in the paper and the writing needs to be improved. I encourage the authors to use the feedback to strengthen the paper and resubmit to another conference in the future.